# Stress-Induced Premature Senescence Related to Oxidative Stress in the Developmental Programming of Nonalcoholic Fatty Liver Disease in a Rat Model of Intrauterine Growth Restriction

**DOI:** 10.3390/antiox11091695

**Published:** 2022-08-29

**Authors:** Basile Keshavjee, Valentine Lambelet, Hanna Coppola, David Viertl, John O. Prior, Laurent Kappeler, Jean-Baptiste Armengaud, Jean-Pierre Chouraqui, Hassib Chehade, Paul-Emmanuel Vanderriele, Manon Allouche, Anne Balsiger, Alexandre Sarre, Anne-Christine Peyter, Umberto Simeoni, Catherine Yzydorczyk

**Affiliations:** 1Department of Woman-Mother-Child, Division of Pediatrics, DOHaD Laboratory, University of Lausanne and Lausanne University Hospital, 1011 Lausanne, Switzerland; 2Department of Nuclear Medicine and Molecular Imaging, University of Lausanne and Lausanne University Hospital, 1011 Lausanne, Switzerland; 3INSERM, Centre de Recherche St Antoine, CRSA, Sorbonne Université, 75000 Paris, France; 4IHU-ICAN Institute of Cardiometabolism and Nutrition, Sorbonne Université, 75000 Paris, France; 5Department of Woman-Mother-Child, Pediatric Nutrition and Gastroenterology Unit, University of Lausanne and Lausanne University Hospital, 1011 Lausanne, Switzerland; 6Department of Biomedical Sciences, University of Lausanne, National Center of Competence in Research Kidney, 1011 Lausanne, Switzerland; 7Department Woman-Mother-Child, Neonatal Research Laboratory, Clinic of Neonatology, University of Lausanne and Lausanne University Hospital, 1011 Lausanne, Switzerland

**Keywords:** intrauterine growth restriction, metabolic syndrome, developmental programming, nonalcoholic fatty liver disease, oxidative stress, cellular senescence

## Abstract

Metabolic syndrome (MetS) refers to cardiometabolic risk factors, such as visceral obesity, dyslipidemia, hyperglycemia/insulin resistance, arterial hypertension and non-alcoholic fatty liver disease (NAFLD). Individuals born after intrauterine growth restriction (IUGR) are particularly at risk of developing metabolic/hepatic disorders later in life. Oxidative stress and cellular senescence have been associated with MetS and are observed in infants born following IUGR. However, whether these mechanisms could be particularly associated with the development of NAFLD in these individuals is still unknown. IUGR was induced in rats by a maternal low-protein diet during gestation versus. a control (CTRL) diet. In six-month-old offspring, we observed an increased visceral fat mass, glucose intolerance, and hepatic alterations (increased transaminase levels, triglyceride and neutral lipid deposit) in male rats with induced IUGR compared with the CTRL males; no differences were found in females. In IUGR male livers, we identified some markers of stress-induced premature senescence (SIPS) (lipofuscin deposit, increased protein expression of p21^WAF^, p16^INK4a^ and Acp53, but decreased pRb/Rb ratio, foxo-1 and sirtuin-1 protein and mRNA expression) associated with oxidative stress (higher superoxide anion levels, DNA damages, decreased Cu/Zn SOD, increased catalase protein expression, increased nfe2 and decreased keap1 mRNA expression). Impaired lipogenesis pathways (decreased pAMPK/AMPK ratio, increased pAKT/AKT ratio, SREBP1 and PPARγ protein expression) were also observed in IUGR male livers. At birth, no differences were observed in liver histology, markers of SIPS and oxidative stress between CTRL and IUGR males. These data demonstrate that the livers of IUGR males at adulthood display SIPS and impaired liver structure and function related to oxidative stress and allow the identification of specific therapeutic strategies to limit or prevent adverse consequences of IUGR, particularly metabolic and hepatic disorders.

## 1. Introduction

The prevalence of metabolic syndrome (MetS) has increased worldwide [1]. It includes obesity, particularly abdominal body fat accumulation, impaired glucose metabolism, dyslipidemia, and arterial hypertension [2,3]. Non-alcoholic fatty liver disease (NAFLD), which is defined as excess fat (>5% weight or volume) deposition in the liver in the absence of excessive alcohol intake, has been identified as the hepatic manifestation of MetS [4]. Nowadays, NAFLD is considered to be the major cause of chronic liver disease worldwide [5].

MetS is mainly explained by insulin resistance [6]; however, environmental factors, such as a sedentary lifestyle, junk food and alcohol consumption have been identified as major contributors to MetS [7,8]. Despite health policies based on physical exercise promotion, reduced calorie intake, and healthy food consumption, the incidence and prevalence of MetS are still increasing worldwide. This phenomenon can partly be explained by the fact that adverse events in the perinatal period can increase the susceptibility to developing cardiometabolic diseases in adulthood. The Developmental Origins of Health and Disease (DOHaD) concept proposed that a suboptimal environment during a critical period of development early in life can induce permanent changes in the structure and function of specific organs that may increase the risk of MetS later in life. In particular, individuals born following intrauterine growth restriction (IUGR), whose low birth weight (LBW) represents an indirect marker, are at risk of developing metabolic disorders in adulthood, such as type-2 diabetes [9], obesity [10], MetS [11] and NAFLD [11,12,13,14,15,16].

The fetal insulin hypothesis proposed that insulin secretion and resistance, genetically determined, can affect intrauterine growth and explain the association between lower birthweight and type-2 diabetes later in life [17]. However, the mechanisms involved in the development of hepatic disorders in individuals born after IUGR are still not well established.

Oxidative stress and cellular senescence have been associated with the pathophysiology of metabolic disorders, such as adiposity, insulin resistance [18] and NAFLD [5,19]. Patients with MetS exhibit increased oxidative damage, as identified by decreased antioxidant defenses, such as reduced superoxide dismutase activity, and an increase in malondialdehyde levels, protein carbonyl and xanthine oxidase activity [20,21,22]. Additionally, total body fat and waist circumference have been positively associated with oxidative stress [23]. An accumulation of senescent cells has been observed in metabolic alterations. In humans, macro- and micro-vasculopathies associated with MetS have been observed with aging [24,25,26]. Indicators of cellular senescence have been observed in patients with NAFLD [27,28,29], as well as in liver alterations induced by transient postnatal overfeeding in a mouse model [30].

Oxidative stress and cellular senescence are also observed in fetal growth restriction [31,32,33]. Increased malondialdehyde levels [34], urinary 8-oxo-7,8 dihydro-deoxyguanosine, and plasma protein carbonylation, but decreased total antioxidant capacity, have been mentioned in the blood sample of pregnant women with growth-restricted fetuses [35], all of which are also consistent with similar observations in the cord blood of IUGR neonates [35,36,37]. Regarding cellular senescence, placentas complicated by fetal growth restriction displayed short telomeres and suppression of telomerase activity [38,39]. Our research team has previously observed the impaired functionality of endothelial colony-forming cells (ECFCs) associated with stress-induced premature senescence (SIPS) isolated from cord blood of LBW newborns [40] and from bone marrow in a rat model of IUGR [41]. However, whether SIPS and oxidative stress can be associated with the developmental programming of NAFLD in individuals born after IUGR is still unknown. 

In the same rat model of IUGR induced by a maternal low-protein (LP) diet during gestation, as previously studied, we investigate, in this manuscript, whether IUGR rats develop metabolic and hepatic alterations at adulthood and whether these dysfunctions are associated with SIPS and oxidative stress.

More particularly, we explore, at adulthood, some metabolic parameters, such as glucose tolerance, body composition, and hepatic structure and function (lipogenesis pathway). We investigate some parameters related to SIPS (lipofuscin staining, pro- and anti-senescence factors) and related to oxidative stress (superoxide anion production, DNA double-strand breaks and antioxidant defenses). In addition, the hepatic structure and some markers of SIPS and oxidative stress are also considered at birth.

## 2. Materials and Methods

### 2.1. Animal Model

The animal model used has been previously described [41]. A rat model of IUGR was induced by an altered maternal diet during gestation. Pregnant rats (Sprague Dawley) were randomly allocated to a control diet (20% casein; CTRL group) throughout gestation or to an isocaloric LP diet (8% casein; IUGR group) (Table 1). Male and female pups in the IUGR group displayed lower birth weights than males and females in the CTRL group, and the difference persisted at 6 months of age [41]. At birth, the litters were equalized to ten pups in both groups to ensure a standardized nutrient supply. In the postnatal period, sample animals were randomly selected within litters for both groups. At 6 months of age, the animals were euthanized by an intraperitoneal injection of pentobarbital (Esconarkon, Streuli Pharma AG, Uznach, Switzerland) at a dose of 150 mg/kg of body weight followed by exsanguination. Every sample animal presented in this study originated from a separate litter.

### 2.2. Intraperitoneal Glucose Tolerance Test (IPGTT)

Intraperitoneal glucose tolerance test (IPGTT) was performed to assess glucose metabolism. Rats were fasted for 16 h. Blood was sampled from the saphenous vein. Blood glucose levels were quantified using a glucose meter (Accu-Chek, Aviva, Roche Diabetes Care SA, Rotkreuz, Switzerland) before, and 15, 30, 60, 90 and 120 min after. the intraperitoneal administration of 20% (2 g/kg) dextrose (Sigma–Aldrich, Saint Louis, MO, USA).

### 2.3. Daily Food Consumption

Each animal was placed in a separate cage. For 7 days, food intake was determined each day by weighing the amount of food remaining from the previous day’s allowance. Animal body weights were measured, and the amount of food intake was reported with respect to the body weight.

### 2.4. Liver Enzyme Measurement

Aspartate aminotransferase (ASAT) and alanine aminotransferase (ALAT) were measured in the CTRL and IUGR groups at 6 months of age, in 16-h fasting plasma samples collected during exsanguination, by the clinical chemistry department of the CHUV (Lausanne, Switzerland).

### 2.5. Body Composition Measurement

At 6 months of age, rats were anesthetized with isoflurane (induction at 5% and maintenance at 3%) and placed prone on the animal bed. Acquisitions were performed using micro-PET/SPECT/CT from Albira Bruker (Bruker Switzerland AG, Fällanden, Switzerland) with a longitudinal field of view of 173.5 mm and 1000 projections, resulting in a voxel size of 125 µm, an X-ray voltage of 45 kV and a current of 200 µA. After reconstruction, resulting tomographs were analyzed using PMOD 3.6 software (PMOD Technologies, Zürich, Switzerland). We selected a slice passing through the inferior articular facet of the 6th lumbar vertebrae. Due to their difference in X-ray attenuation, fat and lean mass can be segmented from each other. A segmentation range for total adipose tissue (TAT) was set between −500 and −100 Hounsfield Units (HUs). To segregate subcutaneous (SAT) from visceral adipose tissue (VAT), semi-automated tools were used to demarcate SAT. The VAT surface was obtained by substituting SAT for TAT. Indeed, SAT is much more easily delineable under the skin than VAT scattered between the abdominal organs. Then, VAT can be expressed as a percentage of the total slice surface.

### 2.6. Liver Analyses

Livers from the CTRL and IUGR groups were collected at birth and at 6 months of age and fixed in formaldehyde to perform histological analyses (hematoxylin/eosin to evaluate the liver structure). Additionally, livers were frozen in liquid nitrogen at 6 months of age, and stored at −80 °C for subsequent investigations.

### 2.7. Steatosis Detection

Steatosis was detected using red oil staining in frozen livers section (from the CTRL and IUGR groups) of 6-month-old rats. Pictures were taken using an inverted microscope (Eclipse Ti2 Series-Nikon Europe B.V., Amsterdam, The Netherlands) by a single examiner (C.Y.). Quantification was performed using ImageJ software, and a “stack image” and a color threshold were applied to identify the stained structure. Results are reported as a red-stained area percentage among the total area [30].

### 2.8. Superoxide Anion Production Evaluation

A superoxide anion (O_2_^•−^) was detected using chemiluminescence. Liver O_2_^•−^ production was evaluated in the CTRL and IUGR males at birth and at 6 months of age as previously described [30,41]. Deparaffinized liver sections (5-μm thick) were stained with hydroethidine (2 μM, Sigma–Aldrich) and incubated in a light-protected humidified chamber at 37 °C for 30 min. The sections were rinsed and mounted using Fluoromount g mounting medium with 4′6-diamidino-2-phenylindole (DAPI; Life Technologies Europe B.V, Zug, Switzerland). A negative control was established through incubation without hydroethidine. Images were obtained blindly using an inverted fluorescent microscope (Eclipse Ti2 Series-Nikon) by a single examiner (C.Y.). Fluorescence was evaluated with ImageJ software, and liver autofluorescence was subtracted.

### 2.9. Oxidative DNA Double-Strand Break 

Liver sections from CTRL and IUGR males at birth and 6-month-old rats were stained with 53BP-1 (1/100, Abcam, Cambridge, UK) overnight at 4 °C. Sections were then washed with PBS and incubated for two hours with Alexa Fluor-647-conjugated goat anti-rabbit IgG (1/200, Abcam). Sections were then rinsed with PBS and mounted using Fluoromount g mounting medium with DAPI. A negative control was established through incubating only with secondary antibody. Slides were observed blindly using a fluorescence microscope (Nikon, Eclipse Ti2 Series) by the same experimenter (C.Y.) [30]. Fluorescence was evaluated with ImageJ software, and liver autofluorescence was subtracted

### 2.10. Histological Detection of Cell Senescence

The presence of lipofuscin, a highly oxidized insoluble protein that accumulates in the cytoplasm, was identified using Sudan Black B (SBB) and Fontana Masson staining only at 6 months of age. Pictures were quantified using ImageJ software, and a “stack image” and a color threshold were applied to identify the stained structure. Results are reported as a red stained area percentage among the total area [30,42]. Additionally, the autofluorescence of liver lipofuscin was detected using an inverted fluorescent microscope (Eclipse Ti2 Series-Nikon) at birth and 6 months of age. Lipofuscin exhibits broad-spectrum autofluorescence [43,44]. A GFP filter set was applied with an exposure time of 400 ms throughout the observation. Pictures were taken by a single examiner (C.Y.).

To localize lipofuscin deposits at 6 months of age, hepatocytes were stained with cytokeratin 18 (1/100, Abcam) overnight at 4 °C. Sections were then washed with PBS and incubated for two hours with Alexa Fluor-647-conjugated goat anti-rabbit IgG (1/200, Abcam). Sections were then rinsed with PBS and mounted using Fluoromount g mounting medium with DAPI. A negative control was established through incubation only with secondary antibody. Slides were observed blindly using a fluorescence microscope (Nikon, Eclipse Ti2 Series) by the same experimenter (C.Y.).

### 2.11. PCR-Selected cDNA Subtraction Kit

Total RNA was extracted with RNAzol reagent, according to the manufacturer’s instructions (Life Technologies Europe B.V., Zug, Switzerland). Total RNA extracted from liver samples at 6 months of age, using the RNeasy Mini kit (Qiagen, Hilden, Germany), was quantified using a Nanodrop (Life Technologies Europe B.V.) and was reverse-transcribed using the PrimeScriptTM RT reagent kit, according to the manufacturer’s instructions (Takara Bio Inc., Shiga, Japan). PCR was performed on a Corbett Rotor-Gene 6000 apparatus with the Rotor-Gene SYBR Green PCR kit (QIAGEN, Hilden, Germany), following the manufacturer’s protocol. The program was as follows: 40 cycles of 5 s denaturation at 95 °C and 10 s annealing/amplification at 60 °C. Primer sequences that were used are summarized in Table 2.

### 2.12. Protein Expression Measurement Using Western Blotting

Liver proteins from the CTRL and IUGR groups were extracted at 6 months of age (from the medial lobe of the snap-frozen liver), as previously described [30,42]. Denatured (10 min at 70 °C) liver proteins (20 μg) from CTRL and IUGR groups were separated on the same gradient gel (NuPAGE 4–12% Bis-Tris gel, Life Technologies Europe B.V.) and transferred overnight at 4 °C to Whatman nitrocellulose membranes (Life Technologies Europe B.V.). Ponceau staining (Life Technologies Europe B.V.) confirmed the presence of membrane proteins. All primary antibody incubations were performed in blocking buffer (TBS-Tween 1%-bovine serum albumin (BSA) 3%) overnight at 4 °C. Antibodies against phospho-AMP-activated protein kinase (pAMPK), AMPK, phospho-protein kinase B (Ser473) (pAKT), AKT, Cu/Zn superoxide dismutase (SOD), catalase, sirtuin-1, FoxO1, retinoblastoma tumor suppressor protein (Rb) and phospho-Rb (Ser807/811) (pRb), p21^WAF^, p53 and acetyl-p53 (Lys382), p16^INK4a^, sterol regulatory element-binding protein 1 (SREBP1), peroxisome proliferator activated receptor gamma (PPARγ), and β-actin from Cell Signaling Technology (Danvers, MA, USA) were purchased and used at the dilutions recommended for immunoblotting (1:1000). Incubations with anti-mouse or anti-rabbit secondary antibodies (1/2000; Cell Signaling, Danvers, MA, USA) were performed for 1 h at room temperature in blocking buffer (TBS-Tween 1%-BSA 3%). Antibodies were visualized using enhanced chemiluminescence Western blotting substrate (Life Technologies Europe B.V.). A G-BOX Imaging System (GeneSys, Syngene, Cambridge, UK) was used to detect specific bands, and each band optical density was measured using specific software (GeneTools 4.03.05.0, Syngene, Cambridge, UK) for all blots.

### 2.13. Statistical Analyses 

All data were presented as mean ± standard deviation (SD). After checking the normal distribution, experimental observations were analyzed using a Student’s *t*-test. GraphPad Prism 9 (version 9.1.0 (221), La Jolla, CA, USA) was used for statistical analyses and creating graphics. The significance level was set at *p* < 0.05.

## 3. Results

### 3.1. Intrauterine Growth Worsened Glucose Intolerance and Increased Visceral Fat Mass 

At 6 months of age, we observed an increase of the area under the curve of glucose measurement (AUC) in IUGR males (+39.48%, *p* < 0.001) compared to the CTRL males (Figure 1A). No differences were observed between CTRL and IUGR females (Figure 1A). Daily food consumption was similar between IUGR and CTRL in males and females (Figure 1B). On the computed tomography (CT) scans, visceral fat mass was increased in IUGR males (+26.68%; *p* < 0.05) compared to the CTRL males (Figure 1C). No differences were found between CTRL and IUGR females (Figure 1C).

### 3.2. Intrauterine Growth Restriction Leads to Liver Function and Structure Alterations

We observed that, at 6 months of life, the liver/body weight ratio was increased in IUGR males (+15%; *p* < 0.05) compared to the CTRL males (Figure 2A). Additionally, IUGR males displayed increased ASAT and ALAT levels (+74%; *p* < 0.01 and +38%; *p* < 0.001, respectively) compared to the CTRL males (Figure 2B,C). No differences were observed concerning the liver/body weight ratio or transaminase expression between IUGR and CTRL in females (Figure 2A–C).

Concerning the hepatic structure, hematoxylin/eosin (H/E) staining identified the presence of hepatocytes with macro-vesicular steatosis, characterized, notably, by rounded, rarefied cytoplasm in livers from IUGR versus CTRL males (Figure 3). No difference was observed between CTRL and IUGR males at birth (Appendix A), as well as in females at 6 months of age (Figure 3). Using red oil staining, we observed an increased deposition of neutral lipids and triglycerides in livers from IUGR males (+5400%; *p* < 0.001) (Figure 4). No accumulation was observed in females (Figure 4).

Some molecular pathways related to lipogenesis (Figure 5) were explored using Western blot in CTRL and IUGR male livers. In IUGR male livers, we observed a decreased pAMPK/AMPK ratio (−42%; *p* < 0.01) (Figure 5A), but an increased expression of the pAKT/AKT ratio (+165%; *p* < 0.05) (Figure 5B), PPARγ (via Western blot: +171%—*p* < 0.01; via RT-qPCR: +377%—*p* < 0.05) (Figure 5C,E) and SREBP1 (also known as SREBF1) (via Western blot: +220%—*p* < 0.01; via RT-qPCR: +303%—*p* < 0.05) (Figure 5D,E).

We also measured some inflammatory markers, such as tumor necrosis factor (TNF)-alpha and nuclear factor kappa B (NFkB), using RT-qPCR in the liver from CTRL and IUGR males at 6 months of age. We observed no significant difference in the mRNA expression of TNF-alpha and NFkB between the two groups (Figure 6).

### 3.3. Oxidative Stress Was Observed in the IUGR Male Livers at Adulthood

Oxidative stress was evaluated in CTRL and IUGR male livers. Using hydroethidine staining, we observed increased superoxide anion production in IUGR compared with the CTRL males (+107%; *p* < 0.001) (Figure 7). DNA double-strand breaks were evaluated via 53BP-1 staining in CTRL and IUGR male livers. We observed an increase in 53BP-1 staining in IUGR male livers (+138%; *p* < 0.001) compared with those from the CTRL group (Figure 8). At birth, we observed no difference in superoxide anion production or in DNA double-strand breaks between CTRL and IUGR males (Appendix A).

We measured antioxidant factor expression (Figure 9) and observed decreased Cu/Zn SOD expression (via Western blot: −50%—*p* < 0.001; via RT-qPCR: −22%—*p* < 0.05) (Figure 9A,C) and increased catalase expression only via Western blot (+23%; *p* < 0.01), but no difference was observed using RT-qPCR (Figure 9B,C), normalized to β-actin, in the livers of IUGR compared with those of CTRL males. In addition, using RT–qPCR, we measured the mRNA expression of nfe2 and keap1. We observed increased nfe2 (+204%; *p* < 0.05) and decreased keap1 (−28%; *p* < 0.05) expressions, normalized to β-actin, in the livers of IUGR compared to the CTRL males (Figure 9D).

### 3.4. Stress-Induced Premature Senescence Was Observed in IUGR Males

Hepatic senescence was detected by lipofuscin staining using SBB, and Fontana Masson, and by autofluorescence in IUGR versus CTRL males at 6 months of age (Figure 10). We identified that lipofuscin deposits were increased in IUGR (+2706%; *p* < 0.001 using SBB staining and +481%; *p* < 0,001 using Fontana Masson staining) compared to the CTRL males and were mainly located in hepatocytes (Figure 11). At birth, no lipofuscin staining was observed in IUGR compared to the CTRL males (Appendix A).

We measured the expression of several markers related to cellular senescence at 6 months of age: pro-senescence factors: p21^WAF^, p53, Ac-p53, p16^INK4a^, and pRb/Rb; anti-senescence factors: sirtuin-1 and FoxO1. In the livers of IUGR versus CTRL rats, we observed, via Western blot, an increased protein expression of p21^WAF^ (+105%; *p* < 0.05) (Figure 12A), p16^INK4a^ (+88%; *p* < 0.05) (Figure 12C) and Acp53 (+89%; *p* < 0.01) (Figure 12E). No difference was observed concerning p53 expression between IUGR and CTRL males (Figure 12D). In contrast, a decreased pRb/Rb ratio (−30%; *p* < 0.01) (Figure 12B), as well as FoxO1 (−71%; *p* < 0.05) (Figure 12F) and sirtuin-1 (−54%; *p* < 0.05) (Figure 12G) protein expressions, were observed in IUGR vs. CTRL male livers. Using RT–qPCR, we identified a decreased gene expression of foxo-1 (−55%; *p* < 0.05) and sirtuin-1 (−99%; *p* < 0.01), but increased p16 (+423%; *p* < 0.05) and p21 (+597%; *p* < 0.05) normalized to β-actin, (Figure 12I) in IUGR males livers compared to those of CTRL males.

## 4. Discussion

Findings from this study demonstrated a sexual dimorphism in the developmental programming of metabolic alterations at adulthood. Increased glucose AUC and visceral fat mass, with some criteria related to NAFLD, were observed at 6 months of age only in males born following IUGR. In addition, we demonstrated that these liver dysfunctions in IUGR males were associated with some markers of SIPS related to oxidative stress. No differences were observed between IUGR and CTRL males at birth.

We have previously observed that only IUGR males had increased systolic blood pressure at 6 months of age [41]. In this study, we performed metabolic investigations at the same age in both sexes. Compared with the CTRL group, only IUGR males displayed an increased AUC after glucose challenge, indicating glucose intolerance, and suggesting that birth weight was inversely correlated with the glucose AUC in males. A reduction in intrauterine growth has been strongly linked to impaired glucose tolerance [9]. Moreover, hyperinsulinemia has been observed in IUGR males in adulthood [45]. Adipose tissue is not uniformly accumulated in the body but is distributed into subcutaneous (SAT) and visceral adipose tissue (VAT), which are highly correlated with each other. However, VAT appears to be the most accurate predictor of cardiometabolic risk [46,47], as it may be seen as a unique endocrine fat depot releasing excess inflammatory cytokines, adipokines and free fatty acids into the portal vein [48]. Increased adiposity in adulthood has been observed in several animal models of IUGR induced by altered maternal nutrition during gestation, placental uterine ligation, or exposure to glucocorticoids [49,50,51,52]. At 6 months of life, non-invasive imaging methods, such as computed tomography, instead of sacrificing the animal to quantify adipose tissue, enable valuable longitudinal assessment [53]. We observed increased visceral fat mass without increased body weight only in IUGR males, possibly because no difference in daily food consumption was observed between IUGR and CTRL males. In fact, in animal models of IUGR, an increase in food intake has often been associated with catch-up growth, leading to obesity in adulthood [54,55,56,57]. Indeed, postnatal catch-up growth was shown to increase adiposity rather than muscle and skeletal growth [58,59]. Concerning the absence of metabolic alterations at 6 months of age in females, it has been shown that female growth-restricted offspring, compared to males of the same age, are normotensive at 6 months of age, but, subsequently, develop increased blood pressure and metabolic alterations, such as increased fat mass, at 12 months of age [60]. This sexual dimorphism has been observed in several animal models of developmental programming [61,62,63]. 

Increased visceral fat mass, glucose intolerance, and arterial hypertension have been associated with NAFLD. It has been shown that the liver is a sexually dimorphic organ, and, principally, male individuals display more severe stages of NAFLD [64]. In growth-restricted male livers at 6 months of life, we identified some parameters related to NAFLD: an increased liver to body weight ratio; the presence of hepatocytes with macro-vesicular steatosis, characterized, notably, by rounded, rarefied cytoplasm and increased triglycerides and neutral lipid deposits, detected via Oil Red O staining. In addition, increased plasmatic transaminases ASAT and ALAT levels were observed in IUGR males. These enzymes are identified as non-invasive indicators of NAFLD [65]. Similar observations have been made in a rat model fed a high-fat and -cholesterol diet [66]. No difference was observed in any of these parameters between IUGR and CTRL females. At birth, we observed no alteration between IUGR and CTRL males in liver H/E staining.

The liver comprises a heterogenous tissue and, in addition to hepatocytes, Kupffer cells have been identified to contribute to liver dysfunctions. These cells are believed to be the main source of pro-inflammatory cytokines, such as TNF alpha [67], which activates NFkB and enhances liver injury, including fibrosis and carcinogenesis [68]. Using RT-qPCR, we observed no difference in the mRNA expressions of TNF alpha and NFkB between IUGR and CTRL males, suggesting that Kupffer cells did not contribute to the liver dysfunctions observed in our IUGR rat model.

The liver is known to play a key role in protein, lipid and glucose homeostasis. It is susceptible to damage, particularly hepatocyte senescence, which has been linked to both acute and chronic liver diseases. Cellular senescence is defined as a decline in cell division capacity and ability to proliferate [69]. Two types of cellular senescence have been identified: replicative senescence, an irreversible phenomenon, identified by a decline in telomere length with each cell cycle [70], and SIPS, initiated in young cells via different mechanisms, including oxidative stress, which can be reversed. It has been shown that the inadequate removal of highly oxidized proteins induces the formation of insoluble protein aggregates, such as lipofuscins, which are commonly observed in aged hepatocytes [71,72,73]. In addition, lipofuscin staining has been proposed as a senescence biomarker, particularly for SIPS [74]. In IUGR 6-month-old male livers, we observed lipofuscin accumulation [75,76], as previously observed in a mouse model with metabolic and hepatic alterations induced by transient postnatal overfeeding [30], where lipofuscin deposits were mainly localized in hepatocytes. We measured the expression of some markers related to cellular senescence. Livers from IUGR males displayed increased p21^WAF^ and p16^INK4a^ protein expressions and decreased pRb/Rb expression. Decreased sirtuin-1 expression has been reported with age, probably due to the lower NAD+ availability [77]. We observed a decreased expression (mRNA and protein) of sirtuin-1 in IUGR male livers. Sirtuin-1 can interact with p53 activity. In fact, sirtuin-1 deacetylates in the C-terminal regulatory domain of p53 [78] and can then inhibit SIPS [79]. In agreement with the decreased sirtuin-1 expression, we observed an increased protein expression of acetylated p53 at Lys-382, a target of sirtuin-1 deacetylase, in IUGR male livers, indicating an impaired deacetylation action of sirtuin-1 and an overall reduction in sirtuin-1 activity [30], which could explain the presence of SIPS in IUGR male livers. As we observed no accumulation of lipofuscin deposit at birth, we did not explore these molecular markers.

There is evidence that cellular senescence, especially in hepatocytes, may modulate fat accumulation in patients with NAFLD [69,80]. We identified alterations of the molecular pathways involved in lipogenesis in IUGR compared to the CTRL males. AMPK is a sensor of the intracellular energy status that coordinates several metabolic pathways, including hepatic lipid metabolism [81]. A strong association has been observed between the reduction in AMPK activity and the incidence of metabolic diseases, such as obesity, diabetes, and NAFLD [82]. AMPK is inactivated in senescent cells [83] in both obese rodents and human subjects [84,85]. Additionally, hepatic AMPK activation is substantially attenuated in NAFLD [86,87], as we observed in IUGR male livers. AMPK activity may be inversely correlated with the activation of AKT. We observed an increased pAKT/AKT expression in IUGR male livers. In AKT-deficient cells, elevated AMPK activity has been observed, whereas cells expressing activated AKT have been observed to reduce AMPK activity [88]. In addition, hepatic de novo lipogenesis can be controlled through a lipogenic transcription factor, SREBP1, which can mediate the expression of lipogenesis-associated triglyceride synthesis and accumulation [89]. SREBP1 can induce increased triglyceride liver accumulation, leading to the development of NAFLD [90], and its expression is increased in the case of cellular senescence [91]. We observed the increased protein expression of SREBP1 in IUGR male livers. The overexpression of SREBP1 in liver tissue cells accelerated triglyceride accumulation in hepatocytes [92,93], which may be due to the activation of AKT [94,95,96,97,98]. In addition, the role of SREBP1 in lipogenesis has been improved by another factor, PPAR-γ. In obese patients with NAFLD who underwent subtotal gastrectomy, a higher protein expression of liver PPAR-γ has been shown, which may reinforce the lipogenic actions associated with SREBP1 upregulation. IUGR male livers displayed a higher protein expression of PPAR-γ than those of CTRL males, which could explain the increase in p16^INK4a^ expression. In fact, it has been shown that PPAR-γ activation accelerates cellular senescence by inducing p16^INK4a^ expression [99]. Sirtuin-1 may also exert a protective role against hepatic steatosis. Indeed, sirtuin-1 overexpression has been shown to provide protection against high-fat-induced hepatic steatosis in mice via the upregulation of gene expression, enhancing fatty acid oxidation and the downregulation of lipogenic gene expression [100,101,102]. Therefore, decreased sirtuin-1 expression and activity in IUGR male livers might have modulated the expression of these lipogenic genes and induced the liver steatosis we observed. 

Several factors can induce SIPS and, notably, oxidative stress [103], which is involved in the pathogenesis of NAFLD [104]. Oxidative stress is defined as an imbalance between reactive species, notably, reactive oxygen species (ROS) and antioxidant defenses. Excessive ROS production can interact with cellular macromolecules, leading to lipid peroxidation, DNA damage and/or induced protein and nucleic acid modifications [105]. Moreover, cellular senescence as a result of tissue injury can induce DNA damage. Notably, p53-binding protein 1 (53BP1) is, in addition to yH2AX, another DNA damage-response protein that binds quickly to DNA damage sites, especially double-strand brakes. We observed an increased 53BP1 staining in the livers of IUGR males compared to the CTRL, as previously found in a mouse model with metabolic and hepatic alterations induced by transient postnatal overfeeding [30]. At birth, we observed no DNA damage. Using hydroethidine staining, we observed increased superoxide anion production in IUGR male livers at 6 months of age only, but not at birth. Decreased Cu/Zn SOD and increased catalase expression may explain the accumulation of superoxide anions because Cu/Zn SOD cannot correctly catalyze the dismutation of superoxide anions to hydrogen peroxide and O_2_. The increased catalase expression may represent a compensatory mechanism to fight against oxidative stress. Antioxidant defenses can be regulated via the interaction between sirtuin-1 and FoxO1 [106,107]. Sirtuin-1 has been shown to upregulate the deacetylation of FoxO, which activates antioxidant defenses to resist oxidative stress [108,109]. The FoxO protein family is widely involved in cell signal transduction, growth and development, apoptosis, and antioxidant stress. FoxO1 and FoxO3 are the most common members of the FoxO family. FoxO1 can regulate the expression of sirtuin-1 by binding to its gene promoter region [110], which creates an autoregulatory feedback loop that regulates sirtuin-1 expression. Chronic stress conditions are likely to downregulate sirtuin-1 levels. In addition to decreased sirtuin-1, we observed the decreased protein and mRNA expression of foxo1 in IUGR male livers compared with the CTRL males, which may explain the decreased Cu/Zn SOD protein expression. Transcription nuclear factor erythroid 2 (nfe2)-related factor 2 (nrf2) plays a key role in cell protection against oxidative stress [111,112,113,114] and is often induced by stressful conditions. We observed increased nfe2 mRNA expression in the livers of IUGR compared with the CTRL male livers, as was found in mice models fed with a high-fat diet containing 42% fat calories from milk [115] or 36% calories derived from soybean oil [116]. Increased hepatic nrf2 expression was also described in a mouse model of diet-induced obesity and fatty liver disease (C57BL/6J mice fed a 12-week high-fat diet) [116]. A cytosolic inhibitor (inrf2), which is also known as kelch-like ech-associated protein 1 (keap1), negatively regulates nrf2 activity by promoting the proteasomal degradation of nrf2 [117]. Under quiescent conditions, nrf2 interacts with keap1 and remains in the cytoplasm, which maintains a low expression of nrf2-regulated genes. In the case of oxidative stress exposure, notably, ROS, nrf2 is released from keap1, trans-locates to the nucleus, and transactivates the expression of cytoprotective genes that improve cell survival. The release of nrf2 from keap1 repression induces the nuclear accumulation of nrf2, suggesting that the nrf2–keap1 system acts as a sensor for oxidative stress. We observed a decreased keap1 mRNA expression in IUGR male livers. Nrf2 may act as a potential lipid metabolic regulator via the activation of PPAR-γ and SREBP1 [115]. Thus, the increased expression of nfe2 might also explain the increased expression of PPAR-γ and SREBP1, leading to fatty liver disease, as observed in IUGR male livers. 

## 5. Conclusions/Perspectives

### 5.1. Conclusions

In this study, we demonstrated a sexual dimorphism in the developmental programming of metabolic disorders at adulthood: only IUGR males displayed glucose intolerance and had increased visceral fat mass. Furthermore, IUGR males had impaired liver structure and function with increased transaminase levels, increased deposition of neutral lipids and triglycerides and altered lipogenesis pathways. At birth, we observed no difference between CTRL and IUGR male livers concerning the hepatic structure, or the markers of SIPS and oxidative stress, so these data suggest that SIPS, related to oxidative stress, could be associated with the impaired hepatic structure and function we observed. 

### 5.2. Limitations/Perspectives

This study was performed on both sexes. As only males developed metabolic and hepatic alterations at 6 months of age, cellular senescence, lipogenesis and oxidative stress investigations were only performed on males. This is consistent with the fact that male individuals display more severe stages of NAFLD. However, it would be interesting to study females at 12 months of age, to observe whether they develop similar alterations. Males should also be studied earlier, at 2 or 4 months of life, to observe whether IUGR livers display some alterations associated with SIPS and oxidative stress, in order to determine whether these mechanisms precede, or are consequences of, impaired liver structure and function. 

Additionally, as we observed modifications of some mRNAs regulating lipogenesis, oxidative stress and SIPS, it would be interesting to explore the expression of some miRNAs regulating these pathways.

In therapeutical approaches, sirtuin-1 plays beneficial roles, especially in regulating hepatic lipid metabolism, oxidative stress and delaying SIPS. As we observed decreased mRNA and a reduced protein expression of sirtuin-1 in IUGR male livers, modulating sirtuin-1 expression using resveratrol may be an interesting approach. In fact, resveratrol was found to improve sirtuin-1 activation, prevent oxidative stress, delay SIPS and protect against cardiometabolic disorders [1,40]. We recently administered treatment using resveratrol to ECFCs isolated from the same animal model presented in this study, and observed a reversion of ECFC dysfunctions in the IUGR group [41] (unpublished data). Therefore, interesting future approaches would be to test whether the above-mentioned metabolic and hepatic alterations could be prevented via in vivo supplementation with a molecule such as resveratrol. 

As regards clinical implications, this study allows identification of specific therapeutic strategies, to limit or prevent adverse consequences of IUGR, particularly metabolic and hepatic disorders.

## Figures and Tables

**Figure 1 antioxidants-11-01695-f001:**
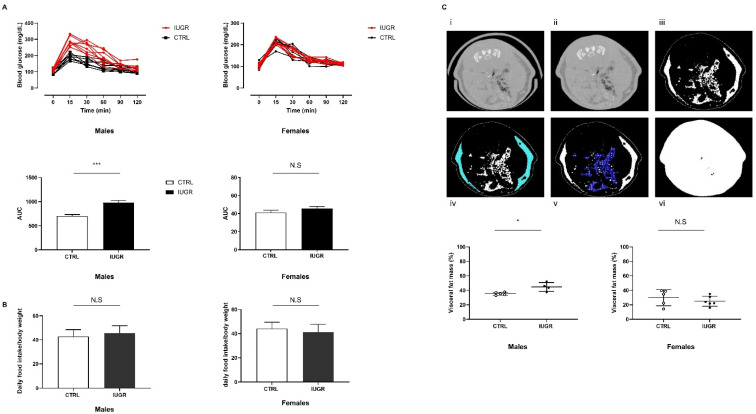
Metabolic explorations in IUGR and CTRL males and females at 6 months of age. AUC after glucose challenge (**A**) and daily food intake (**B**) was determined in *n* = 8 animals/group/sex. VAT measurement was performed in males and females (**C**); CT of a slice passing through the 6th lumbar vertebra before (i) and after (ii) rat holding structure removal. TAT segmentation expressed in white (iii) and SAT demarcation (iv). VAT scattered between the organs (v) expressed as a total slice (vi) percentage. *n* = 4–6 animals/group/sex. * *p* < 0.05; *** *p* < 0.001; N.S: not significant.

**Figure 2 antioxidants-11-01695-f002:**
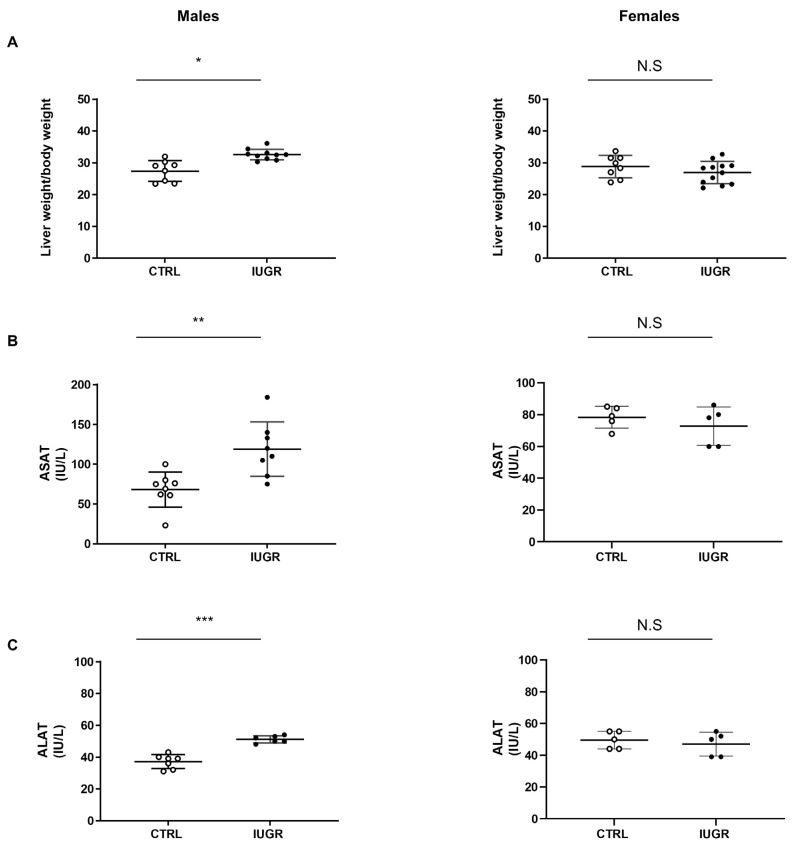
Liver weight and transaminase levels in IUGR and CTRL males and females at 6 months of age. Liver to bodyweight ratio (**A**) (*n* = 8–12 animals/group/sex), plasma levels of aspartate aminotransferase (ASAT) (**B**) and alanine aminotransferase (ALAT) (**C**) (*n* = 5–8 animals/group/sex) were measured in males and females. * *p* < 0.05; ** *p* < 0.01; *** *p* < 0.001; N.S: not significant.

**Figure 3 antioxidants-11-01695-f003:**
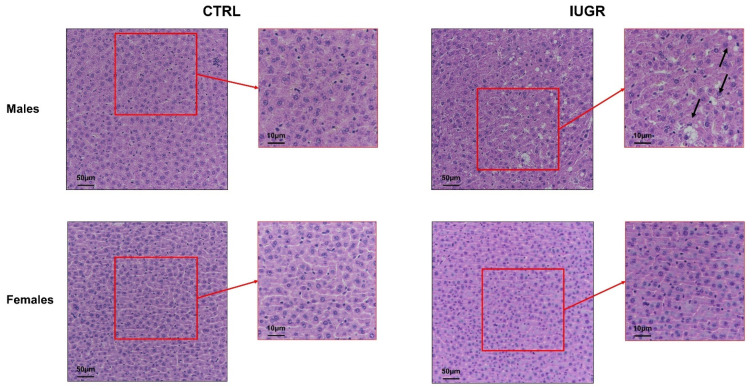
Liver structure in CTRL and IUGR males and females at 6 months of age. Liver structure was evaluated using H/E staining in males and females. The macro-vesicular steatosis is indicated by black arrows. *n* = 5 animals/group/sex. Magnification (20 and 40×). Scale bar = 10 and 50 μm.

**Figure 4 antioxidants-11-01695-f004:**
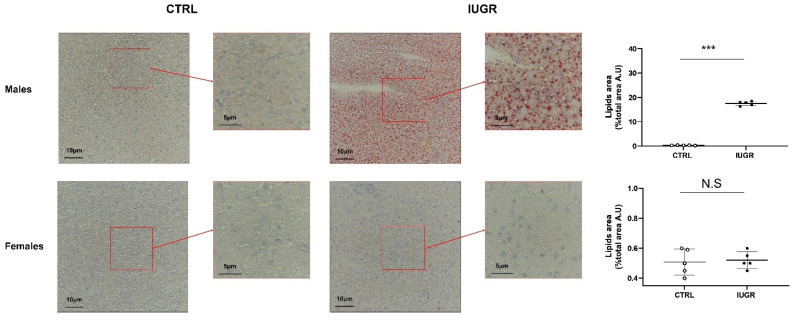
Neutral lipids and triglycerides in the liver from CTRL and IUGR males and females at 6 months of age. Oil Red O staining was performed to identify neutral lipids and triglycerides in males and females. *n* = 5 animals/group/sex. *** *p* < 0.001; N.S: not significant. Magnification (40 and 60×). Scale bar = 5 and 10 μm.

**Figure 5 antioxidants-11-01695-f005:**
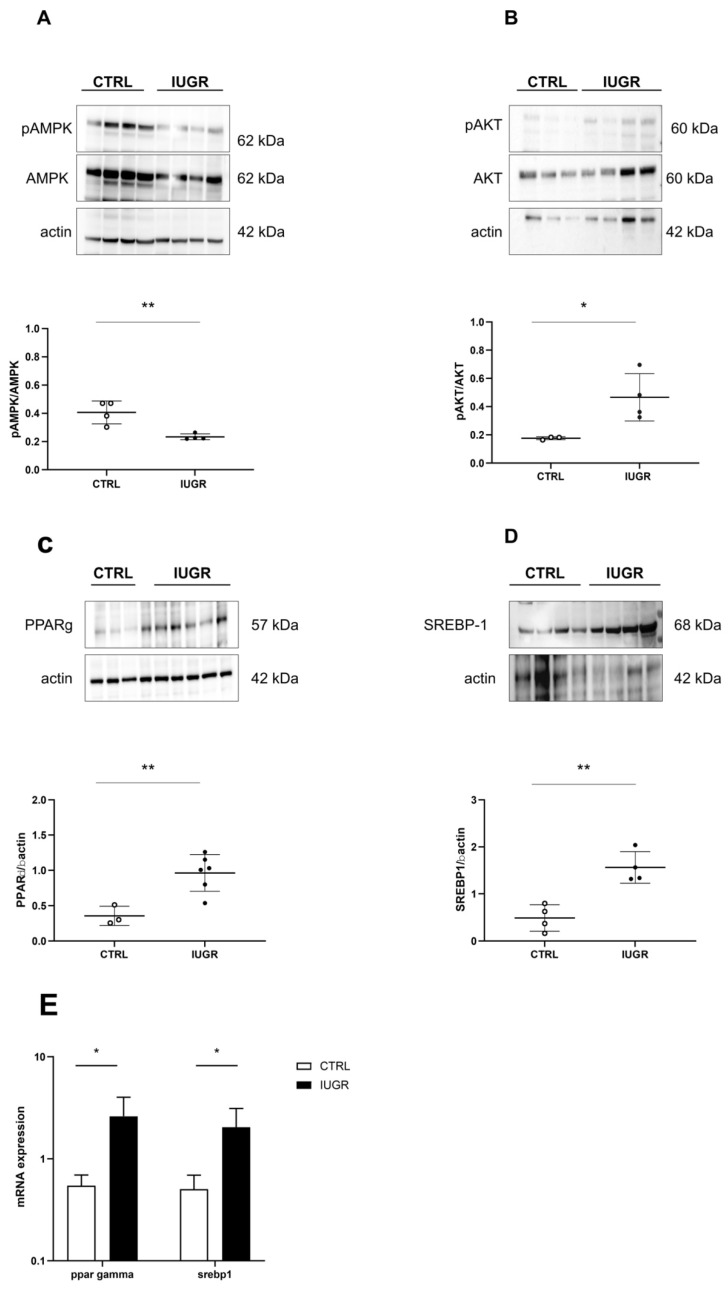
Molecular pathways related to liver lipogenesis in CTRL and IUGR males at 6 months of age. Protein expressions of pAMPK/AMPK (**A**), pAKT/AKT (**B**), PPARγ (**C**), and SREBP1 (**D**) were measured through Western blot. * *p* < 0.05; ** *p* < 0.01; *n* = 3–5 animals/group. The mRNA expression levels of ppar-gamma (**E**) and srebp1 (**E**) normalized to β-actin were determined using RT–qPCR in CTRL and IUGR male livers. * *p* < 0.05; *n* = 5 animals/group.

**Figure 6 antioxidants-11-01695-f006:**
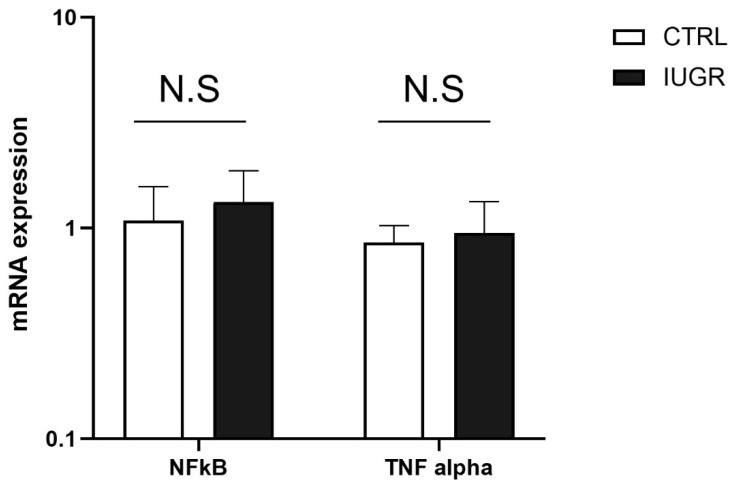
The mRNA expression levels of NFkB and TNF alpha normalized to β-actin were determined using RT-qPCR in CTRL and IUGR male livers; N.S: not significant. *n* = 5 animals/group.

**Figure 7 antioxidants-11-01695-f007:**
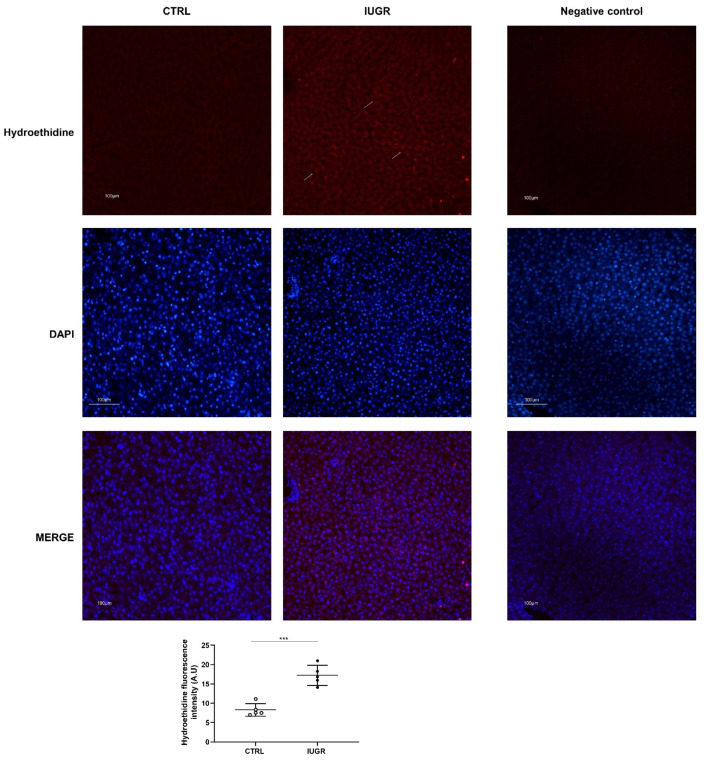
Superoxide anion production in the liver from CTRL and IUGR males at 6 months of age. The superoxide anion production (arrows) was evaluated using hydroethidine staining in CTRL and IUGR males. A negative control was established. Nuclei were counterstained with DAPI. *n* = 5 animals/group. *** *p* < 0.001. Magnification (20×). Scale bar = 100 μm.

**Figure 8 antioxidants-11-01695-f008:**
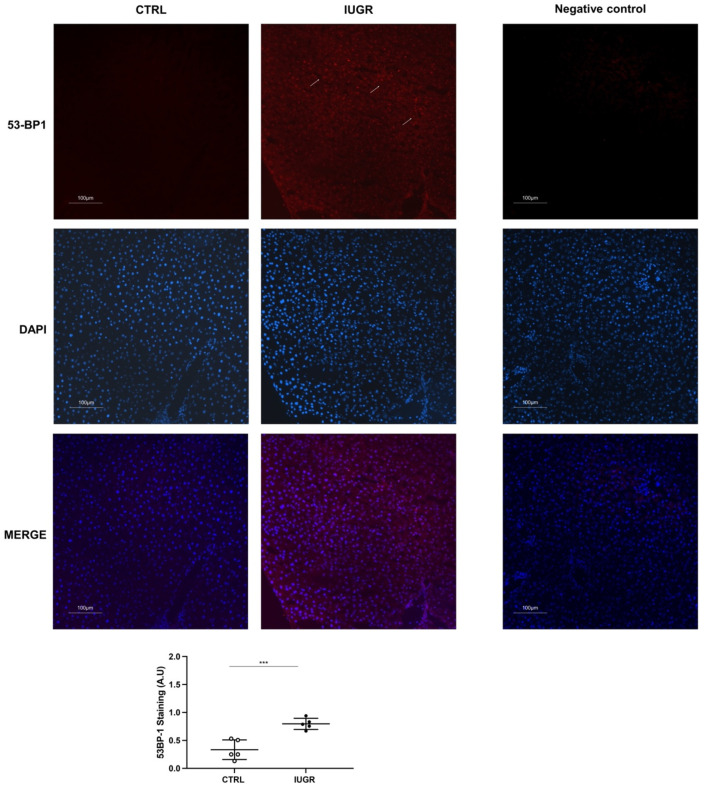
DNA double-strand breaks (arrows) were evaluated using 53BP-1 staining in the liver from CTRL and IUGR males. A negative control was established. Nuclei were counterstained with DAPI. *n* = 5 animals/group. *** *p* < 0.001. Magnification (20×). Scale bar = 100 μm.

**Figure 9 antioxidants-11-01695-f009:**
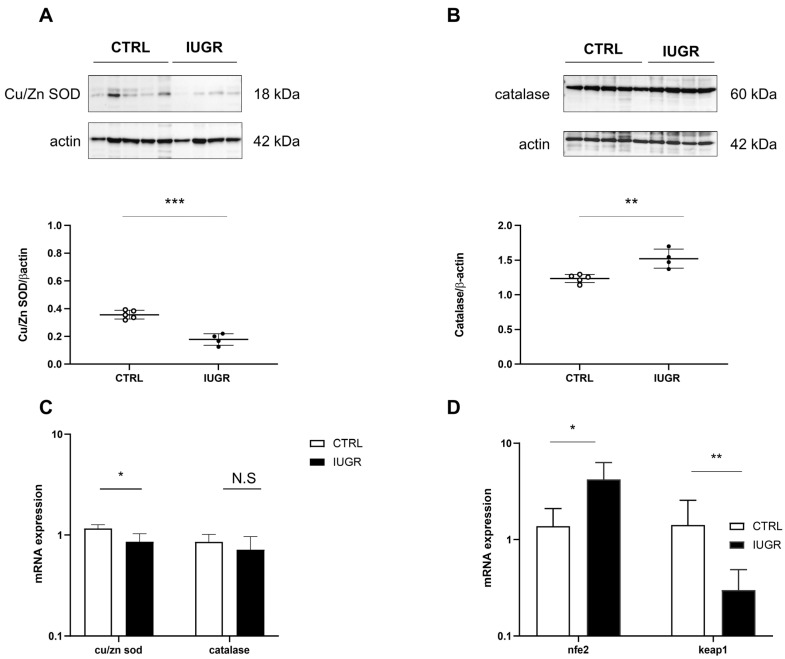
Oxidative stress factors in CTRL and IUGR male livers at 6 months of age. Protein expression levels of Cu/Zn SOD (**A**) and catalase (**B**) were measured using Western blotting in CTRL and IUGR groups; ** *p* < 0.01; *** *p* < 0.001; *n* = 4–5 animals/group. In addition, mRNA expression levels of cu/zn sod, catalase, nfe2 and keap1 normalized to β-actin were determined using RT–qPCR in both groups (**C**,**D**). * *p* < 0.05; ** *p* < 0.01; N.S: not significant; *n* = 5 animals/group.

**Figure 10 antioxidants-11-01695-f010:**
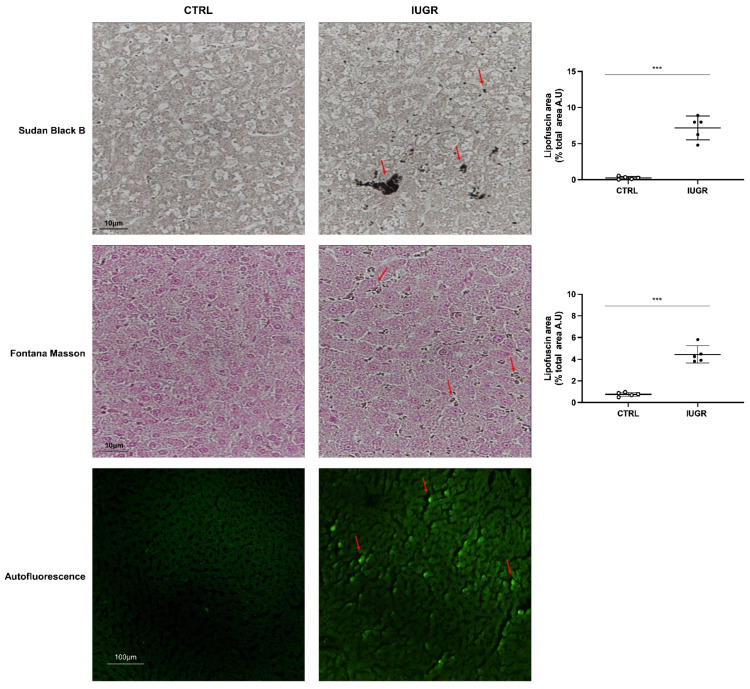
Lipofuscin detection in CTRL and IUGR male livers at 6 months of age. Lipofuscin detection was performed using SBB and Fontana Masson staining and with autofluorescence. Lipofuscin deposits are indicated by arrows. *n* = 5 animals/group. *** *p* < 0.001. Magnification (20 and 40×). Scale bar = 10 and 100 μm.

**Figure 11 antioxidants-11-01695-f011:**
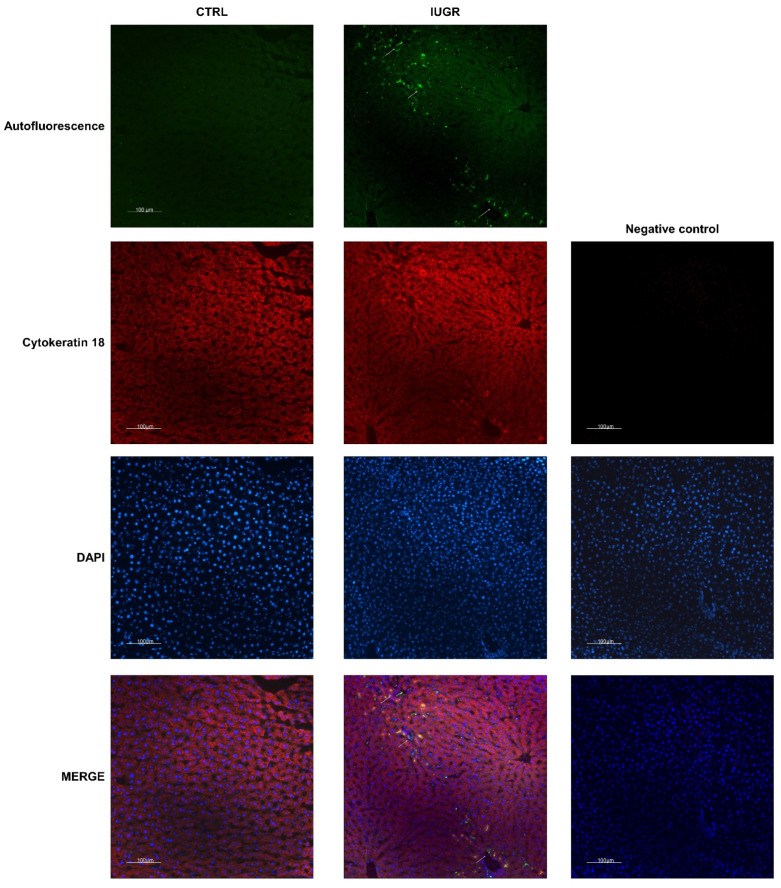
Lipofuscin localization in CTRL and IUGR male livers at 6 months of age. Lipofuscin detection was performed using autofluorescence (GFP filter), and hepatocytes were identified using cytokeratin-18 staining. Lipofuscin deposits are indicated by arrows. A negative control was established. Nuclei were counterstained with DAPI. *n* = 5 animals/group. Magnification (20×). Scale bar = 100 μm.

**Figure 12 antioxidants-11-01695-f012:**
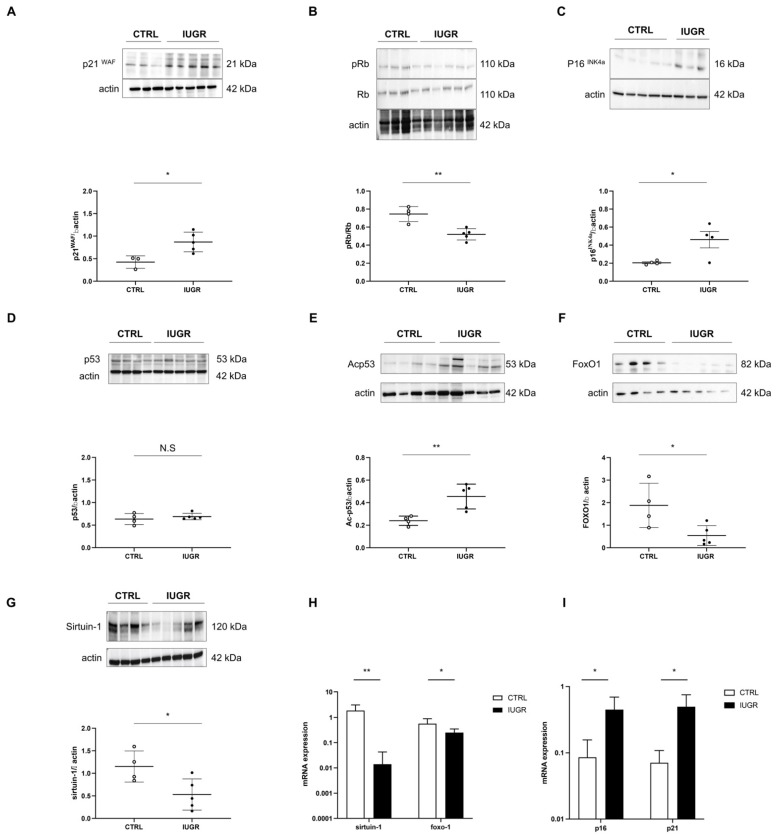
Molecular pathways related to cellular senescence in CTRL and IUGR male livers at 6 months of age. The protein expression of p21^WAF^ (**A**), pRb/Rb (**B**), p16^INK4a^ (**C**), p53 (**D**), Acp53 (**E**), FoxO1 (**F**) and sirtuin-1 (**G**) were measured using Western blotting in the CTRL and IUGR groups. * *p* < 0.05; ** *p* < 0.01; N.S: not significant. *n* = 3–5 animals/group. mRNA expressions of foxo-1 and sirtuin-1 (**H**), p16 and p21 (**I**) normalized to β-actin were measured in the CTRL and IUGR groups using RT–qPCR. * *p* < 0.05; ** *p* < 0.01; *n* = 3–5 animals/group.

**Table 1 antioxidants-11-01695-t001:** Diet composition.

-	CODE	U8959P	U8959P
-	VERSION	1	40
-	**FORMULA INGREDIENTS G/KG**	210 Control	Hypoproteic 8%
61250A99	dextrose	380	380
63400A99	casein	230	90
60366A99	pregelatinized cornstarch	200	345
91101A99	pre-mixture of minerals PM 205B 7%	70	70
64254A99	crude cellulose	60	60
61964A99	lard	30	30
61910A99	colza oil	10	10
61930A99	corn oil	10	10
90513A99	pre-mixture of vitamins PV 200 1% (casein)	10	5
61254A99	sucrose	0	0
65884A99	choline	0	0
65815A99	vitamin e	0	0
65300A99	calcium carbonate	0	0
65805A99	inositol	0	0
65841A99	nicotinic acid	0	0
65806A99	para-amino-acid ben	0	0
65876A99	vitamin b12	0	0
65828A99	vitamin k3 mnb	0	0
65801A99	vitamin a	0	0
65831A99	vitamin b1 thiamin	0	0
65836A99	riboflavin	0	0
65856A99	vitamin b6 pyridoxine	0	0
65851A99	vitamin b5	0	0
65811A99	vitamin d3	0	0
65861A99	folic acid	0	0
65880A99	vitamin c	0	0
65873A99	biotin	0	0

**Table 2 antioxidants-11-01695-t002:** Primer list.

Primers	Forward Primer	Reverse Primer
Actin NM031144.3	AACACCCCAGCCATGTACG	GCATGAGGGAGCGCGTAAC
Foxo1 NM001191846.3	CGTCCTCGAACCAGCTCAA	TTGGCGGTGCAAATGAATAG
nfe2 NM001012224.1	CTAGTTCGGGACATCCGTCG	TCTCGCTCCAACTGCACAAT
keap1 NM057152.2	CTTCGGGGAGGAGGAGTTCT	ATTTGACCCAGTCGATGCAC
sirtuin-1 NM001372090.1	TGTGCAGTGGAAGGAAAGCA	CTGCAACCTGCTCCAAGGTA
Tnf-alpha NM012675.3	CATCCGTTCTCTACCCAGCC	GGGCTCTGAGGAGTAGACGA
Nfκb NM001276711.1	ACCACTGTCAACAGATGGCCC	TTCGCACACGTAGCGGAATC
Srebf1 NM001276707.1	CGCTCTTGACCGACATCGAA	GAGGCCACAGTGCTCATTCT
ppar gamma NM013124.3	TCGCTGATGCACTGCCTATG	TGATTCCGAAGTTGGTGGGC
p 16 NM031550.2	AACACTTTCGGTCGTACCCC	CCCAGCGGAGGAGAGTAGAT
p 21 NM080782.4	ATGTCCGACCTGTTCCACAC	ATCGGCGCTTGGAGTGATAG

## Data Availability

Data Availability Statement: https://doi.org/10.5281/zenodo.5979281 (accessed on 5 February 2022).

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
