# Peer review of "Stress-Induced Premature Senescence Related to Oxidative Stress in the Developmental Programming of Nonalcoholic Fatty Liver Disease in a Rat Model of Intrauterine Growth Restriction"

_antioxidants, 2022, doi:10.3390/antiox11091695_

Round 1

Reviewer 1 Report (New Reviewer)

The manuscript, “Stress-induced premature senescence related to oxidative stress in the developmental programming of nonalcoholic fatty liver disease in a rat model of intrauterine growth restriction” by Keshavjee et. al. uncovers that liver of intrauterine growth restricted (IUGR) males as adults show stress-induced premature senescence (SIPS) is related to oxidative stress associated with impaired liver structure and function. The manuscript is well written, and experiments were well planned and will be of broad interest to the researchers in the area.

However, some poor microscopic images and western blots should be redone for the better believability of the conclusions made from them. Generally speaking, a) the images should be crisp with an enlarged inset. Nuclear staining intensity should not vary between groups along with the markers that are being compared and conclusions being made, b) all figures with microscopic images should be accompanied by graphs quantifying the result as the experiments had 5 biological repeats as described in the Figure legends, c) Western blots where two groups are being compared, should be run on the same blot (at least with 3 samples from each group), and d) the proteins that are being tested for induction should accompany with qPCR for mRNA expression.

For example:

·      Figure 9A, if possible western blot should be run on one gel that contains both the groups with a smaller number of samples. It is always very difficult to compare two gels even though there is the Actin loading control.

·      The immunostaining images in Figure 11 are of poor quality with variable DAPI staining and of high background. Better crisp images must be provided along with graphs after quantification and statistics. As the n is 5, it shouldn’t be difficult.

·      Figure 12 blots are of poor quality, and for 12a, the p21 blot, comparing two groups of samples, needs to be performed on the same gel (like the p16 blot). as the actin bands are unequal and not as clean.

·      qPCR of p21, p16, and p53 would be better suited to this experiment.

Author Response

Journal: Antioxidants

Manuscript ID: 1833509

Type: Article

Title: Stress-induced premature senescence related to oxidative stress in the developmental programming of nonalcoholic fatty liver disease in a rat model of intrauterine growth restriction

Basile Keshavjee, Valentine Lambelet, Hanna Coppola, David Viertl, John O. Prior, Laurent Kappeler, Jean-Baptiste Armengaud, Jean-Pierre Chouraqui, Hassib Chehade, Paul-Emmanuel Vanderriele, Manon Allouche, Anne Balsiger and Alexandre Sarre, Anne-Christine Peyter, Umberto Simeoni and Catherine Yzydorczyk

We would like to thank the reviewers for their valuable remarks. Our manuscript has been carefully revised according to their requests and suggestions.

Please find below our point-by-point responses (in blue) to the reviewers’ comments (in black).

The manuscript, “Stress-induced premature senescence related to oxidative stress in the developmental programming of nonalcoholic fatty liver disease in a rat model of intrauterine growth restriction” by Keshavjee et. al. uncovers that liver of intrauterine growth restricted (IUGR) males as adults show stress-induced premature senescence (SIPS) is related to oxidative stress associated with impaired liver structure and function. The manuscript is well written, and experiments were well planned and will be of broad interest to the researchers in the area.

However, some poor microscopic images and western blots should be redone for the better believability of the conclusions made from them.

Generally speaking, a) the images should be crisp with an enlarged inset. Nuclear staining intensity should not vary between groups along with the markers that are being compared and conclusions being made,

Authors’ response:

All images have been improved as requested.

  1. b) All figures with microscopic images should be accompanied by graphs quantifying the result as the experiments had 5 biological repeats as described in the Figure legends,

Authors’ response:

We have quantified all microscopic images using ImageJ software and we have included the quantification in methods. However, for reasons of sensitivity it was not possible to quantify the autofluorescence images.

  1. c) Western blots where two groups are being compared, should be run on the same blot (at least with 3 samples from each group), and d) the proteins that are being tested for induction should accompany with qPCR for mRNA expression.

Authors’ response:

We have performed new blots with CTRL and IUGR samples on the same gel concerning pAMPK/AMPK, PPARgamma, SREBP1, Cu/Zn SOD, catalase, p21waf and p53.

In addition, we have performed qPCR for ppar gamma, srebp1, cu/zn sod, catalase, p16 and p21.

Reviewer 2 Report (New Reviewer)

This manuscript entitled "Stress-induced premature senescence related to oxidative stress 2 in the developmental programming of nonalcoholic fatty liver 3 disease in a rat model of intrauterine growth restriction" provides an IUGR animal model to elucidate the correlation between IUGR and oxidative stress caused premature senescence. It is an interesting issue for the readers, however, some typo errors occurred in the text body and English need to be modified .

1. For example, in line 85-86, "... urinary 8-oxo-7,8dihydro-20deoxyguanosine...", "20" should be removed.

2. the description should be re-edited to more concise, for example,  line 72-74, "Oxidative stress and cellular senescence have been associated with the pathophysiology of metabolic disorders and NAFLD. Oxidative stress has been associated with adiposity and insulin resistance". the two sentences should be united.

3. please the authors check the English throughout.

4. the resolution of histological photoes should be improved, they are too dim to recognize.

5. regarding to the IUGR males express more severe adiposity and visceral fat, the authors only describes the results and the similarity as compared with the other studies but not discuss the rationale. This reviewer suggests it needs to be revised.

Author Response

Journal: Antioxidants

Manuscript ID: 1833509

Type: Article

Title: Stress-induced premature senescence related to oxidative stress in the developmental programming of nonalcoholic fatty liver disease in a rat model of intrauterine growth restriction

Basile Keshavjee, Valentine Lambelet, Hanna Coppola, David Viertl, John O. Prior, Laurent Kappeler, Jean-Baptiste Armengaud, Jean-Pierre Chouraqui, Hassib Chehade, Paul-Emmanuel Vanderriele, Manon Allouche, Anne Balsiger and Alexandre Sarre, Anne-Christine Peyter, Umberto Simeoni and Catherine Yzydorczyk

We would like to thank the reviewers for their valuable remarks. Our manuscript has been carefully revised according to their requests and suggestions.

Please find below our point-by-point responses (in blue) to the reviewers’ comments (in black).

This manuscript entitled "Stress-induced premature senescence related to oxidative stress 2 in the developmental programming of nonalcoholic fatty liver 3 disease in a rat model of intrauterine growth restriction" provides an IUGR animal model to elucidate the correlation between IUGR and oxidative stress caused premature senescence. It is an interesting issue for the readers, however, some typo errors occurred in the text body and English need to be modified.

  1. For example, in line 85-86, "... urinary 8-oxo-7,8dihydro-20deoxyguanosine...", "20" should be removed.

Authors’ response:

We have removed “20”

  1. the description should be re-edited to more concise, for example,  line 72-74, "Oxidative stress and cellular senescence have been associated with the pathophysiology of metabolic disorders and NAFLD. Oxidative stress has been associated with adiposity and insulin resistance". the two sentences should be united.

Authors’ response:

We have unified the sentences and made the paragraph more consistent.

Modifications in the revised manuscript:

“Oxidative stress and cellular senescence have been associated with the pathophysiology of metabolic disorders, such as adiposity, insulin resistance [18] and NAFLD [5, 19]. Patients with MetS exhibit increased oxidative damage, as identified by decreased antioxidant defenses, such as reduced superoxide dismutase activity, and an increase in malondialdehyde levels, protein carbonyl and xanthine oxidase activity [20-22]. Additionally, total body fat and waist circumference have been positively associated with oxidative stress [23]. An accumulation of senescent cells has been observed in metabolic alterations. In humans, macro- and microvasculopathies associated with MetS have been observed with aging [24-26]. Indicators of cellular senescence have been observed in patients with NAFLD [27-29], as well as in liver alterations induced by transient postnatal overfeeding in a mouse model [30].

Oxidative stress and cellular senescence are also observed in fetal growth restriction [31-33]. Increased malondialdehyde levels [34], urinary 8-oxo-7,8 dihydro-deoxyguanosine, and plasma protein carbonylation, but decreased total antioxidant capacity, have been mentioned in pregnant women with growth-restricted fetuses [35], all of which are also consistent with similar observations in IUGR neonates [35-37]. Concerning cellular senescence, pregnancies complicated by fetal growth restriction displayed short telomeres and suppression of telomerase activity [38, 39]”.

  1. please the authors check the English throughout.

Authors’ response:

The English in the entire of manuscript has been checked by MDPI's Author Services (please, see the attached English-Editing-Certificate).

  1. the resolution of histological photoes should be improved, they are too dim to recognize.

Authors’ response

All the histological images have been improved and quantified. However, for reasons of sensitivity it was not possible to quantify the autofluorescence images.

  1. regarding to the IUGR males express more severe adiposity and visceral fat, the authors only describes the results and the similarity as compared with the other studies but not discuss the rationale. This reviewer suggests it needs to be revised.

Authors’ response

We have discussed this point in the revised manuscript.

Modifications in the revised manuscript:

“Adipose tissue is not uniformly accumulated in the body and is distributed into subcutaneous (SAT) and visceral adipose tissue (VAT), which are highly correlated with each other. However, VAT appears to be the most accurate predictor of cardiometabolic risk [46, 47] as it may be seen as a unique endocrine fat depot releasing excess inflammatory cytokines, adipokines and free fatty acids into the portal vein [48]. Increased adiposity in adulthood has been observed in several animal models of IUGR induced by altered maternal nutrition during gestation, placental uterine ligation, or exposure to glucocorticoids [49-52]. At 6 months of life, non-invasive imaging methods, such as computed tomography, instead of sacrificing the animal to quantify adipose tissue, enable valuable longitudinal assessment [53]. We observed increased visceral fat mass without increased body weight only in IUGR males, possibly because no difference in daily food consumption was observed between IUGR and CTRL males. In fact, in animal models of IUGR, an increase in food intake has often been associated with a catch-up growth, leading to obesity in adulthood [54-57]. Indeed, postnatal catch-up growth was shown to increase adiposity rather than muscle and skeletal growth [58, 59]”.

  1. Fox, C. S., J. M. Massaro, U. Hoffmann, K. M. Pou, P. Maurovich-Horvat, C. Y. Liu, R. S. Vasan, J. M. Murabito, J. B. Meigs, L. A. Cupples, R. B. D'Agostino, Sr., and C. J. O'Donnell. "Abdominal Visceral and Subcutaneous Adipose Tissue Compartments: Association with Metabolic Risk Factors in the Framingham Heart Study." Circulation 116, no. 1 (2007): 39-48.
  2. Armani, A., A. Berry, F. Cirulli, and M. Caprio. "Molecular Mechanisms Underlying Metabolic Syndrome: The Expanding Role of the Adipocyte." FASEB J 31, no. 10 (2017): 4240-55.
  3. Nielsen, S., Z. Guo, C. M. Johnson, D. D. Hensrud, and M. D. Jensen. "Splanchnic Lipolysis in Human Obesity." J Clin Invest 113, no. 11 (2004): 1582-8.
  4. Desai, M., and C. N. Hales. "Role of Fetal and Infant Growth in Programming Metabolism in Later Life." Biol Rev Camb Philos Soc 72, no. 2 (1997): 329-48.
  5. McMillen, I. C., and J. S. Robinson. "Developmental Origins of the Metabolic Syndrome: Prediction, Plasticity, and Programming." Physiol Rev 85, no. 2 (2005): 571-633.
  6. Hales, C. N., M. Desai, and S. E. Ozanne. "The Thrifty Phenotype Hypothesis: How Does It Look after 5 Years?" Diabet Med 14, no. 3 (1997): 189-95.
  7. Seckl, J. R., and M. J. Meaney. "Glucocorticoid Programming." Ann N Y Acad Sci 1032 (2004): 63-84.
  8. Sasser, T. A., S. E. Chapman, S. Li, C. Hudson, S. P. Orton, J. M. Diener, S. T. Gammon, C. Correcher, and W. M. Leevy. "Segmentation and Measurement of Fat Volumes in Murine Obesity Models Using X-Ray Computed Tomography." J Vis Exp, no. 62 (2012): e3680.
  9. Desai, M., D. Gayle, J. Babu, and M. G. Ross. "Programmed Obesity in Intrauterine Growth-Restricted Newborns: Modulation by Newborn Nutrition." Am J Physiol Regul Integr Comp Physiol 288, no. 1 (2005): R91-6.
  10. Desai, M., D. Gayle, J. Babu, and M. G. Ross. "Permanent Reduction in Heart and Kidney Organ Growth in Offspring of Undernourished Rat Dams." Am J Obstet Gynecol 193, no. 3 Pt 2 (2005): 1224-32.
  11. Coupe, B., V. Amarger, I. Grit, A. Benani, and P. Parnet. "Nutritional Programming Affects Hypothalamic Organization and Early Response to Leptin." Endocrinology 151, no. 2 (2010): 702-13.
  12. Yousheng, Jia, T. Nguyen, M. Desai, and M. G. Ross. "Programmed Alterations in Hypothalamic Neuronal Orexigenic Responses to Ghrelin Following Gestational Nutrient Restriction." Reprod Sci 15, no. 7 (2008): 702-9.
  13. Khandelwal, P., V. Jain, A. K. Gupta, M. Kalaivani, and V. K. Paul. "Association of Early Postnatal Growth Trajectory with Body Composition in Term Low Birth Weight Infants." J Dev Orig Health Dis 5, no. 3 (2014): 189-96.
  14. Carolan-Olah, M., M. Duarte-Gardea, and J. Lechuga. "A Critical Review: Early Life Nutrition and Prenatal Programming for Adult Disease." J Clin Nurs 24, no. 23-24 (2015): 3716-29.

Round 2

Reviewer 1 Report (New Reviewer)

The manuscript was revised, and questions were answered for the most part. A few minor things as described below need to be corrected.

1. In the Introduction, the paragraph before the last: The authors should mention whether the defects are seen in the mother's uterus or the fetuses?

2. Insets for the microscopic images might help (Inset means a small square area from the image in question is magnified and shown right beside the original image.)

3. Some beta-actin blots are blurry.

4. The Western blots in figure 9 are overexposed (oversaturated). A lower exposure from this experiment and quantification from that will be preferred).

Author Response

This manuscript is a resubmission of an earlier submission. The following is a list of the peer review reports and author responses from that submission.

Round 1

Reviewer 1 Report

This manuscript is an improvement over the previous one, with much more data. However, it is still not clear what signals contribute to the molecular mechanisms that the authors claim. In other words, there is a lack of evidence to show that it is the cause of NAFLD, and direct involvement needs to be demonstrated by inhibition experiments or other means.

In addition, Western blotting data should be sufficiently clean, and mTORC1 is a complex, so the description pmTORC1 is inappropriate. Further modifications would be needed in this regard as well.

Reviewer 2 Report

This study focuses on the investigation of

This study focuses on the investigation of mechanisms of developmental programming of adult liver diseases specifically, non alcoholic liver fatty disease (NFALD). Using the model if intrauterine growth retardation (IUGR) and methods of molecular biology, biochemistry, histology, experimental physiology, immunohistochemistry, immunoblotting and RT-PCR, that Authors provided evidence that IUGR induces long-term liver pathological changes. Specifically, they found liver hypertrophy and the induction of markers of senescence, oxidative damage, glucose intolerance and disturbed lipogenesis. Importantly, these results are in good agreement with the data indicating that neonatal growth retardation caused by Cryptosporidial gastroenteritis increases long-term and irreversible hepatocyte ploidy and liver hypertrophy.  

This is a well done study that took a lot of work. It can be published.   There is just one small point.   In the Result section where you analyse senescent associated genes, please, differ between pro-senescent genes and anti-senescent genes (Sirt1, Foxo1).